# Preoperative Risk Stratification of Increased MIB-1 Labeling Index in Pituitary Adenoma: A Newly Proposed Prognostic Scoring System

**DOI:** 10.3390/jcm11237151

**Published:** 2022-12-01

**Authors:** Ivan Maiseyeu, Ági Güresir, Hartmut Vatter, Ulrich Herrlinger, Albert Becker, Johannes Wach, Erdem Güresir

**Affiliations:** 1Department of Neurosurgery, University Hospital Bonn, 53127 Bonn, Germany; 2Division of Clinical Neurooncology, Department of Neurology and Centre of Integrated Oncology, University Hospital Bonn, 53127 Bonn, Germany; 3Department of Neuropathology, University Hospital Bonn, 53127 Bonn, Germany

**Keywords:** pituitary adenoma, MIB-1, score, progression-free survival

## Abstract

The MIB-1 index is an important risk factor for progression-free survival (PFS) in pituitary adenoma (PA). Preoperatively, the MIB-1 index is not available in the decision-making process. A preoperative method regarding MIB-1 index estimation in PA has not been evaluated so far. Between 2011 and 2021, 109 patients with tumor morphology data, MIB-1 index data, and inflammatory and pituitary hormone laboratory values underwent surgery for PA. An MIB-1 index cutoff point (≥4/<4%) determines the probability of PFS in completely resected PA. An elevated MIB-1 index (≥4%) was present in 32 cases (29.4%) and was significantly associated with increased IGF-1, age ≤ 60, increased ACTH, and increased fibrinogen levels in the multivariable analysis. A scoring system (“FATE”) using preoperative IGF-1, age, ACTH, and plasma fibrinogen level enables the estimation of the MIB-1 index (sensitivity 72%, specificity 68%). The FATE score is also significantly associated with the time to PA progression after the complete resection of the PA. We propose the FATE score to preoperatively estimate the risk of an elevated MIB-1 index (≥4%), which might enable tailoring to medical decision-making, and follow-up interval scheduling, as well as inform future studies analyzing proliferative activities.

## 1. Introduction

Pituitary adenomas (PAs) are considered to be predominantly benign neoplastic diseases and account for 15% of all central nervous system (CNS) tumors [1]. The prediction of the natural course of PAs is very challenging because of a vast heterogeneity of influencing factors such as clinical signs, neuropathological characteristics, proliferative activity, and the growth pattern with potential invasion into surrounding anatomical structures [2]. The reduction of mass effect to relieve clinical signs (e.g., loss of vision) and the prevention of such dysfunctions, in case of further tumor growth, are the primary indications for surgical resection. Due to the low-risk profile of the transsphenoidal approach, neurosurgical resection is the treatment of choice in all cases except for prolactinoma. 

Elevated cellular proliferation is the main avenue of oncogenesis [3]. The Molecular Immunology Borstel (MIB-1)-1 labeling index is an established immunohistochemical diagnostic to identify nuclear structures that are found in cells during proliferation. The Ki-67 antigen can be detected in the nuclei of neoplastic cells in the following phases of mitosis and cell division cycle: G1, S, and G2. Hence, the labeling of this antigen is a simple method to identify the growing fraction of tumor tissue [4,5,6]. Moreover, the MIB-1 labeling index was suggested to distinguish PAs and pituitary carcinomas [7]. Elevated MIB-1 labeling indices were reported to be associated with a higher probability of PA progression [8]. However, the routine examination and inclusion of the MIB-1 labeling index for the classification of PAs are still highly debated. A systematic review analyzing 28 studies on the MIB-1 labeling index, identified 18 studies that found increased MIB-1 labeling indices in recurrent PAs, whereas 10 studies found no correlation [9]. 

To achieve an optimum long-term outcome regarding PA progression, reliable preoperative evaluation, adequate medical information about the goals of surgical therapy, and surgical treatment with preservation of function are of paramount importance. Nevertheless, the MIB-1 labeling index is not available in the preoperative phase of the oncological decision-making process and interactive doctor-patient consultation. An increased MIB-1 labeling index implies an increased need to achieve maximum cytoreductive surgical therapy. To date, there are approaches such as machine learning models using T2 weighted magnetic resonance imaging (MRI) or nomograms combining clinical, demographic, and imaging characteristics to preoperatively estimate the MIB-1 labeling index [10,11]. However, no simple and quick-to-use scoring sheet exists so far to enable an estimation of the MIB-1 index prior to surgical resection. Hence, we have investigated our patient population of sporadic PAs regarding preoperative clinical signs, laboratory inflammatory and hormone markers, and imaging risk factors for an elevated MIB-1 labeling index. Moreover, the present investigation was intended to devise a proposal for a new scoring sheet to display demographic data, laboratory inflammatory data, and endocrine functioning to identify PA patients who are at risk of having an elevated MIB-1 labeling index.

## 2. Materials and Methods

### 2.1. Study Design and Inclusion Criteria

Between January 2011 and December 2021, 269 patients were surgically treated for PAs at the neurosurgical institution. A retrospective review of PA patient data was performed. The criteria for inclusion in this investigation were neuropathologically confirmed primary PA, age at diagnosis ≥ 18 years, the availability of the MIB-1 indices, preoperative plasma and serum inflammatory markers (fibrinogen and C-reactive protein (CRP)), pituitary hormone levels, and neurosurgical treatment via a microscopic resection. PA patients with multiple endocrine neoplasia type 1 (MEN1)-associated PA were excluded due to different treatment regimens and different neuropathological characteristics [12,13]. One hundred and nine patients were included in the final data analysis (see Figure 1).

### 2.2. Data Recording

The following clinical data were retrospectively recorded and summarized in a computerized database file (SPSS, version 27, IBM Corp., Armonk, NY): age at diagnosis, sex, comorbidities, Karnofsky Performance Status (KPS), body mass index (BMI), maximum tumor size (mm), tumor growth pattern (cavernous sinus invasion), presence of pituitary apoplexy, WHO classification based on postoperative histopathological examination, immunohistochemistry, the extent of tumor resection based on postoperative gadolinium-enhanced magnetic resonance imaging (MRI), and postoperative follow-up data. Preoperative MR imaging was routinely performed within 2 days before PA surgery. PA size was calculated using a diameter-based method selecting the single largest diameter on a single transverse preoperative contrast-enhanced T1-weighted MR plane [14]. Laboratory information system Lauris (version 17.06.21, Swisslab GmbH, Berlin, Germany) was used for the recording of laboratory values. Venous blood samples were drawn within one day prior to PA surgery. The individual blood draws were conducted at constant time points, which enables an investigation of the time to PA progression. The routine testing included the following parameters: complete blood count, kidney and liver testing, coagulation profile, plasma fibrinogen, and serum c-reactive protein (CRP). Preoperatively, plasma fibrinogen concentrations were calculated using the Clauss method, which involves the administration of a standard and high concentration of thrombin (Dade^®^ thrombin reagent, Siemens Healthineers, Erlangen, Bavaria, Germany) to platelet-poor plasma. Afterward, fibrinogen concentration is investigated using a reference curve. CRP concentrations were measured by turbidimetric immunoassays with a CRPL3 reagent (Roche, Basel, Switzerland) [15]. Moreover, a panel to determine pituitary hormone levels (Thyroid-stimulating hormone (TSH), Insulin-like growth factor 1 (IGF-1), Adrenocorticotropic hormone (ACTH), Follicle-Stimulating Hormone (FSH), Luteinizing hormone (LH), prolactin, estradiol, testosterone, and growth hormone) was also routinely collected within this examination.

### 2.3. Neuropathology

The neuropathological classification was performed according to the 2016 WHO criteria [16]. All neuropathological diagnoses underwent renewed investigation to reconfirm that diagnoses were in line with those requirements. Immunohistochemical labeling was carried out, as previously described before, for paraffin-embedded biopsy tumor samples [17,18]. The MIB-1 labeling index was calculated using the following antibody kit: Anti-Ki67 (Clone 2B11+PD7/26). Diaminobenzidine was used for visualization, and a neuropathological investigation had been carried out by an expert neuropathologist (AB). Further immunohistochemistry for each adenohypophyseal cell lineage and the detection of p53 was performed.

### 2.4. Follow-Up Scheduling

Clinical and imaging follow-up (MR imaging) appointments were performed 3 months after surgery as well as annually for the following 5 years. Earlier clinical and imaging examinations were set in case of novel or progressive clinical signs (e.g., visual dysfunction, and clinical signs of hormone dysfunction) deficits as well as radiological signs of PA regrowth. Progression was defined as a 25% increase in the volume of the previous PA size [19]. The time to PA progression was defined as the time interval between the initial surgery and the first subsequent treatment (e.g., radiotherapy or redo surgery). 

### 2.5. Statistical Analysis

Data were organized and analyzed using SPSS for Mac (version 27.0; IBM Corp, Armonk, New York, NY, USA). The dichotomization of the MIB-1 labeling index into normal and elevated groups was performed according to a previously identified cutoff value (<4/≥4%) [20]. Kaplan–Meier charts with the corresponding log-rank test confirmed the significance of the MIB-1 labeling index regarding PA progression in a completely resected PA. Normally distributed variables were reported as mean (+/− standard deviation (SD)). Comparisons of categorical data were performed using Fisher´s exact test (two-sided) and an independent *t*-test for continuous data. Receiver operating characteristic curves were constructed for age, ACTH, IGF-1, FSAH, and fibrinogen to analyze their association with the MIB-1 labeling index. The areas under the ROC curve (AUC) were analyzed with regard to the optimal cutoff values for those variables (CRP, fibrinogen, tumor size). Variables that resulted in a *p*-value <0.10 according to the univariable analysis were included in the multivariable analysis. Multivariable binary logistic regression analysis of preoperative factors influencing the MIB-1 labeling index was performed. Dichotomized variables were investigated using the Wald test. A *p*-value threshold set at <0.05 was defined as statistically significant. Significant variables of the multivariate analysis were included in a five-point scoring system regarding the estimation of the MIB-1 labeling index. The ROC curves and Kaplan–Meier charts of the developed score and its association with PFS were also calculated. 

## 3. Results

### 3.1. Probability of Progression-Free Survival in Pituitary Adenoma and Prognostic Value of MIB-1 Labeling Index in the Prediction of Recurrent Pituitary Adenoma 

MR imaging follow-up was available in all 229 patients (229/269; 85.1%) who underwent surgery for pituitary adenoma between January 2011 and December 2021. A gross total resection was achieved in 147 patients (147/229; 64.2%), and a subtotal resection was performed in 82 of those patients (82/229; 35.8%). The mean time to PA progression in those who underwent a GTR was 112.9 (95% CI: 104.2–121.5) months, and in those who underwent an STR was 92.8 months, respectively (*p* < 0.001). Figure 2A shows the corresponding Kaplan–Meier chart. The diagnostic performance of the MIB-1 labeling index regarding PA progression in those who underwent a GTR was further investigated. MIB-labeling indices were available for 133 patients (133/147; 90.5%) who underwent a GTR with available postoperative follow-up data. The mean time (months) to pituitary adenoma progression in those patients with an elevated MIB-1 labeling index (≥4%, *n* = 30) was 85.5 (95% CI: 71.9–99.0) months, and 118.9 (95% CI: 111.9–125.9) months in those with a normal MIB-1 labeling index (<4%, *n* = 103), respectively (log-rank test result: *p* = 0.029). Figure 2B shows the probabilities of PA progression for gross totally resected tumors stratified by MIB-1 labeling indices groups (≥4/<4%).

### 3.2. Patient Characteristics of the Screening group for Baseline Characteristics Being Associated with Elevated MIB-1 Labeling Index

One hundred and nine patients were surgically treated for pituitary adenoma and imaging, laboratory (inflammatory and hormone), and immunohistochemical data were available. The median age at PA diagnosis was 60 years (IQR 47–70) and there was a male predominance among the study population (49 females (45.0%)) and 60 males (55.0%). The median preoperative Karnofsky performance scale (KPS) at admission was 90 (IQR 90–100). Further characteristics are summarized in Table 1.

### 3.3. Pituitary Adenoma Growth Pattern, Surgical Therapy, and Neuropathological Pituitary Adenoma Types

The median maximum diameter (IQR, in mm) of pituitary adenomas was 21 (14–27). The invasion of the tumor into the cavernous sinus was observed in 43 patients (39.4%). Six patients (5.5%) presented with pituitary apoplexy. Gross totally resected patients with available MIB-1 labeling indices, imaging data, and complete preoperative inflammatory as well as hormone panel was achieved in 67 (61.5%) patients. Neuropathological investigations using immunohistochemistry to determine the adenohypophysial-cell lineage were performed. Gonadotroph adenoma (49/109; 45%) was the most common type of pituitary adenoma in the present study cohort, followed by plurihormonal (32.1%) and null cell adenoma (10.1%). The median (IQR) MIB-1 labeling index was 3% (2–4). The expression of p53 was observed in 44 (40.4%) patients. 

### 3.4. Screening for Associations between MIB-1 Labeling Index and Patient Characteristics

An MIB-1 labeling index ≥ 4% was found in 32 (29.4%) patients and 77 (70.6%) patients had an MIB-1 labeling index < 4%. PA patients with an elevated MIB-1 labeling index were significantly younger compared to those with an MIB-1 index of <4%. The mean baseline ACTH levels (39.54 ± 34.36 vs. 26.76 ± 20.34; *p* = 0.03) were significantly higher in those PA patients with an MIB-1 labeling index ≥4%. Continuous data of mean values of plasma fibrinogen levels (3.55 ± 0.89 vs. 3.19 ± 0.89; *p* = 0.06) and baseline IGF-1 (293.46 ± 310.77 vs. 189.56 ± 238.51; *p* = 0.06) also tended to be significantly higher in patients with an MIB-1 staining index ≥4%. Furthermore, baseline FSH levels were lower in those with an elevated MIB-1 index (≥4%) compared to those with a normal MIB-1 index (8.07 ± 12.63 vs. 13.53 ± 18.47; *p* = 0.09). Tumor size was also not associated with the MIB-1 labeling index. Thirteen patients (16.9%) with a MIB-1 index <4% had a microadenoma, and six patients (18.7%) with a MIB-1 index ≥4% had a microadenoma, respectively (*p* = 0.79). Furthermore, the mean maximum diameter (in mm) of those with an MIB-1 index <4% was 21.8± 9.8, whereas the mean maximum diameter in those patients with an MIB-1 index ≥4% was 21.3 ± 9.0 (*p* = 0.81). The expression of p53 was not associated with the MIB-1 labeling index in our cohort. P53 expression was homogeneously distributed among both MIB-1 labeling indices groups. Immunohistochemically detected plurihormonal PAs were observed in 24 cases (31.2%) among patients with an MIB-1 index <4%, whereas 11 plurihormonal PAs (34.4%) were found among those with an MIB-1 index ≥4% (*p* = 0.82). The mean MIB-1 labeling index among those with a plurihormonal PA was 3.09 ± 1.74, and 3.18 ± 1.76 among those with a non-plurihormonal PA, respectively (*p* = 0.80). Additional demographic, tumor growth pattern, medication, and laboratory features in PA patients were not heterogeneously distributed among the MIB-1 groups and are shown in Table 2. 

ROC curves were constructed and the AUCs of age at diagnosis, plasma fibrinogen, IGF-1, and ACTH in the estimation of an MIB-1 labeling index (≥4%) were performed (Appendix A). The AUCs for age at diagnosis, plasma fibrinogen, IGF-1, ACTH, and FSH were 0.72 (95% CI: 0.61–0.82), 0.64 (95% CI: 0.52–0.75), 0.71 (95% CI: 0.60–0.82), 0.65 (95% CI: 0.53–0.77), and 0.57 (95% CI: 0.45–0.68), respectively. The following optimum cutoff values regarding MIB-1 index (≥4%) estimation were identified: age at diagnosis (≤60/>60), plasma fibrinogen (≥3.15/<3.15 g/L), IGF-1 (≥126.1/<126.1 ng/mL), ACTH (≥25.1/<25.1 pg/mL), and FSH (≤5.75/>5.75 mIU/mL). The corresponding values regarding sensitivity, specificity, and Youden´s index of each variable were determined (see Appendix A). Patients with an increased IGF-1 (≥126.1 ng/mL) had a mean (+/− SD) BMI of 30.7 +/− 8.7, whereas patients with an IGF-1 <126.1 ng/ml had a mean (+/− SD) BMI of 28.2 +/− 4.9 (independent *t*-test result: *p =* 0.08). Acromegaly, because of chronic overproduction of IGF-1 levels, was present in 18 (16.5%) cases. Patients with an acromegaly had a mean (+/− SD) MIB-1 labeling index of 2.9 +/− 1.4, and those without an acromegaly had a mean (+/− SD) MIB-1 labeling index of 3.1 +/− 1.7, respectively (*p* = 0.56). Subanalyzes of associations between IGF-1 with age and sex were performed. It was found from the subanalyzes that there was a significant association between IGF-1 and age but not with sex. Twenty-seven (27/49; 55.1%) female patients had an IGF-1 ≥ 126.1 ng/mL, whereas 23 (23/60; 38.3%) male patients had an IGF-1 ≥ 126.1 ng/ml (Fisher´s exact test (two-sided): *p* = 0.09). Patients with an IGF-1 ≥126.1 ng/mL had a mean age of 50.4 ± 16.2, whereas those with an IGF-1 <126.1 ng/mL had a mean age of 64.2 ± 14.3 (independent *t*-test: *p* < 0.001). Multivariate binary logistic regression analysis with the inclusion of the following variables was performed: age at diagnosis (≤60/>60), plasma fibrinogen (≥3.15/<3.15 g/L), IGF-1 (≥126.1/<126.1 ng/mL), ACTH (≥25.1/<25.1 pg/mL), and FSH (≤5.75/>5.75 mIU/mL). The multivariate analysis revealed that age at diagnosis (≤60/>60), plasma fibrinogen (≥3.15/<3.15 g/L), IGF-1 (≥126.1/<126.1 ng/mL), and ACTH (≥25.1/<25.1 pg/mL) were significantly associated with an MIB-1 labeling index ≥4%. Figure 3 summarizes the results of the multivariate analysis.

### 3.5. Scoring System

We evaluated and devised a scoring system to estimate the MIB-1 labeling in pituitary adenoma. The present score was developed with the following aims: (1) Feasible estimation of the MIB-1 labeling index using routinely recordable preoperative features, and (2) easy inclusion into the clinical workflow. The following allocation of the points in the new score, which we called the “FATE” score, ranging from 0 to 5 points (Figure 4) was applied: Preoperative IGF-1 ≥126.1 ng/mL (2 points); preoperative age at diagnosis ≤60 years (1 point); preoperative ACTH ≥ 25.1 pg/mL (1 point); preoperative plasma fibrinogen ≥ 3.15 g/L (1 point). The mean total score points in patients with an MIB-1 labeling index ≥4% were 3.7 (SD = 2.18), and 2.7 (SD = 1.17) in patients with an MIB-1 labeling index <4%, respectively (*p* = 0.009).

The ROC curve was created and the AUC of FATE score estimating an MIB-1 labeling index ≥ 4% was performed. The AUC for the FATE score in the estimation of an increased MIB-1 labeling index (≥ 4%) was 0.79 (95% CI: 0.70–0.88, *p* < 0.001). Using a threshold set at a total scoring value of 3 points, the score yields a sensitivity of 72.0%, a specificity of 68.0% (Youden´s index: 0.40), a positive predictive value of 48.0%, and a negative predictive value of 85.2%. Figure 5 shows the ROC curve with the corresponding results of the analysis. A total score value of <3 points results in an 85.2% probability of not finding an MIB-1 labeling index ≥ 4%.

### 3.6. FATE Score and Progression-free Survival

The development of the FATE score was primarily carried out to enable a sufficient preoperative estimation of the MIB-1 labeling index. The ROC curve analysis of the FATE score in the prediction of tumor progression after a completely resected pituitary adenoma was performed. The AUC for the FATE score was 0.88 (95% CI: 0.77–0.99). The optimum cutoff threshold of the FATE score predicting PFS was ≥4/<4 points. The sensitivity and specificity of the FATE score using the optimum threshold for predicting PA progression were 100.0% and 75.0%, respectively (Youden´s index: 0.75). Figure 6A displays the ROC curve analysis. We investigated PA progression in the study cohort of patients who underwent gross total resection using a dichotomization of the FATE score into <4 (*n* = 45) vs. ≥4 (*n* = 16) points. The mean (+/−SD) follow-up time was 32.8 +/− 28.3 months.

Three patients (3/16; 18.75%) of those with complete resection and a FATE score ≥4 had a recurrence, whereas no patient who underwent a gross total resection with a FATE score <4 had a PA regrowth. The mean time to PA recurrence in those with a gross total resection and an increased FATE score (≥4) was 94.0 months. The log-rank test found a significantly shorter time to PA progression in gross totally resected patients having a FATE score ≥4 compared to those with a FATE score <4 (*p* = 0.014). Figure 6B displays the Kaplan–Meier curve of PFS in gross totally resected pituitary adenoma stratified by a FATE score of “0–3 points” and “4 or 5 points”. 

## 4. Discussion

Residual tumor volume, invasive growth, and young age are known poor predictors regarding regrowth in sporadic PAs [21]. An increased MIB-1 labeling index is considered a molecular marker for aggressive tumor biology, which implies an increased risk for invasive growth and shortened time to tumor progression in PAs [22]. Several investigations have revealed that an elevated MIB-1 labeling index is associated with increased proliferative activity and aggressive nature in PAs [23]. In a typically elective setting regarding the hospital admission of PA patients, it is essential that PA patients as well as their relatives are provided with a maximally profound physician–patient dialog. Nevertheless, the MIB-1 labeling index reflecting the proliferative activity cannot be used in the preoperative therapy planning regarding the extent of resection or weighing-up of surgical and conservative treatment. The present investigation suggests a score to estimate an elevated MIB-1 labeling index. This system strives to identify patients at risk of a high MIB-1 labeling index and uses four preoperative characteristics. Moreover, this scoring template might enable the identification of high-risk patients regarding a shortened time to PA progression.

Our results can be described as follows: (1) a threshold MIB-1 labeling index value of 4% enables a risk stratification regarding recurring and nonrecurring PAs in those who underwent complete resection; (2) increased IGF-1, young age at diagnosis, increased ACTH, and increased plasma fibrinogen were significantly associated with an elevated (≥4%) MIB-1 labeling index; (3) at least one characteristic among young age, increased plasma fibrinogen, and increased ACTH combined with an increased IGF-1 seems to identify patients at risk of an elevated MIB-1 labeling index; and (4) the presence of at least two variables among young age, increased plasma fibrinogen and increased ACTH in combination with an increased IGF-1 results in an increased risk for shortened time to PA regrowth in completely resected PAs.

In the present investigation, we created a Kaplan–Meier chart for the probability of PFS in completely resected PAs stratified MIB-1 index. The optimum threshold was set at ≥4% based on the literature [20]. We found that an MIB-1 index ≥4% is significantly associated with a shortened time to PA progression in completely resected PAs. Preoperatively, an accurate estimation of the MIB-1 labeling index might guide neurosurgeons, endocrinologists, and radiotherapists to provide a tailored treatment schedule. The MIB-1 labeling index was also integrated into a five-tiered classification system regarding invasive growth and proliferative potential. This mentioned classification system for pituitary adenomas underwent an external validation in four independent cohorts [24].

The insulin-like growth factor-1 is synthesized in the liver and acts as a mediator of GH. Elevated serum concentrations of IGF-1 are broadly accepted as the screening test of choice in the diagnostic workflow of patients with acromegaly [25]. Several studies suggested the MIB-1 labeling index as an important clinical outcome parameter regarding tumor control in somatotroph PAs. However, previous studies found no significant association between the MIB-1 labeling index and IGF-1 serum levels [26]. Epidemiological studies suggested a strong association between circulating serum IGF-1 levels and the risk of several cancers such as breast cancer. Furthermore, IGF-1 signaling is potentially associated with cancer progression [27,28]. In breast cancer, IGF-1 levels were found to strongly correlate with the MIB-1 index [29]. Several potential pathophysiological mechanisms might be responsible for our findings. IGF-1 has multiple functions and different roles in the development and progression of several diseases. In a physiological setting, IGF-1 inhibits apoptotic effects and supports cell survival, whereas, in pathophysiological conditions, IGF-1 can enhance cancer progression or increase the number of adipocytes [30,31,32,33,34]. However, our identified optimum cutoff value of IGF-1 regarding the identification of an elevated MIB-1 index is within the physiological reference interval. Furthermore, the clinical symptom of acromegaly itself was no predictor of an increased MIB-1 labeling index. Therefore, the identified association seems to be paradoxical. Nevertheless, those findings suggest that the created scoring system is not exclusively designed for a cohort of somatotroph PAs, and it might be generally transferrable to all primary PAs. Further investigations with a special focus on the association between IGF-1 and PA progression are necessary.

Furthermore, age was also found to have an inverse association with an increased MIB-1 labeling index. This result is also in line with the findings of Cai et al. [11]. They found that the mean age of patients with an MIB-1 labeling index ≥3% is significantly higher compared to those with an MIB-1 labeling index <3%. This inverse association between age and MIB-1 index was also reconfirmed in further investigations of nonfunctional PAs, and somatotroph PAs [35,36]. Furthermore, advanced age was found to effectively inhibit tumor recurrence in nonfunctioning PAs according to the results of the multivariable Cox regression analysis in a retrospective study of 145 patients [37]. 

We found an association between ACTH levels and increased MIB-1 index. This finding is also supported by the study of Pizzaro et al. [38] which measured the MIB-1 labeling index in 159 PAs and revealed that ACTH-secreting adenomas have significantly higher MIB-1 labeling indices [38]. Moreover, Mastronardi et al. [39] found that ACTH-secreting PAs have a mean MIB-1 labeling index of 5.88 +/− 9.13%, whereas other hormone-secreting or nonfunctioning PAs had a mean MIB-1 labeling index of 2.33 +/− 2.4%.

Plasma fibrinogen was found to be independently associated with the MIB-1 index. Plasma fibrinogen is linked to the interleukin-6 (IL-6) gene promoter and is induced by the autocrine functioning of pituitary adenoma cells [40]. In a pathological study evaluating the localization and expression of IL-6, IL-6 receptor, and the signal-transducing subunit (gp130) using immunohistochemistry and reverse transcription PCR, IL-6 was predominantly expressed in ACTH- and FSH/LH-secreting cells. Furthermore, IL-6 might function in GH and prolactin-secreting cells through paracrine and endocrine pathways, whereas IL-6 may function in FSH-secreting PA cells in an autocrine manner [40]. The analysis of the cytokine secretome in 24 PAs in primary cultures using an immunoassay panel with 42 cytokines found that PAs with a deleterious immune phenotype including dense macrophage infiltrates and a cluster of differentiation (CD) 4^+^ T lymphocytes had higher MIB-1 labeling indices. Hence, it was suggested that PA-derived chemokines might enhance the recruitment of macrophages, neutrophils, and T cells into the tumor tissue resulting in a more aggressive behavior [41]. Hence, the systemic inflammatory burden was also identified to be higher in PA patients compared to healthy individuals [42]. The secretome of PAs also includes hormones such as ACTH which are secreted into the circulation and significantly modulate the hematopoiesis as well as circulating immune cells contributing to the degree of systemic inflammatory burden. This pathophysiological condition of PA’s secretome-induced inflammation is well-known for Cushing´s disease [43,44]. Marques et al. [45] performed a retrospective evaluation of 424 PA patients and investigated the usefulness of blood-based inflammation markers to predict the disease course. They found that a score using serum inflammation markers might predict invasive and refractory PAs. Nevertheless, it has to be reminded that there is also a potential confounding effect because systemic inflammatory markers might be influenced by further comorbidities and corticosteroid treatment. Furthermore, plurihormonal PAs were not associated with an increased MIB-1 index in our cohort. However, it also has to be reminded that the determination methods of plurihormonal PA using clinical signs, serum hormone concentrations, and pathological results are increasingly discussed [46]. In the present series, the plurihormonal PAs are determined using immunohistochemical methods as recommended by other investigators [46,47]. Nevertheless, the immunohistochemical detection of hormones might not always result in a laboratory increase in serum hormone concentrations or clinical endocrine signs. Hence, it is debatable whether this classification has some clinical implications. This phenomenon might be explained by the fact that hormones secreted by the PA are biologically inactive or that they lost their functioning after entry into the blood system [46,48]. Moreover, in the present series, we did not identify an association between p53 expression and an increased MIB-1 labeling index. The prognostic value of determining p53 expression in PAs is controversial and there is no recommendation to routinely include it in the classification workflow according to the WHO [16]. Furthermore, different staining methods and heterogeneous cohorts (proportions of micro- and macroadenomas) also resulted in a broad range from 17% to 60% regarding the frequency of observed p53 expression [49,50]. 

The novel FATE score in the present investigation provides a score to preoperatively estimate an increased MIB-1 labeling index and might guide physicians in estimating the risk of tumor progression in completely resected PAs. This risk index may facilitate the preoperative treatment planning and the patient–physician dialog because neuropathological characteristics are only available after surgery so far. PA patients with an elevated FATE score (≥3) who prefer a conservative regimen of an incidental PA have to be informed about a more stringent follow-up imaging schedule regarding the time intervals. Furthermore, the FATE score was also found to be significantly associated with the probability of PFS in completely resected PA patients. Therefore, the identification of the FATE score as a potential sufficient surrogate marker for the MIB-1 index might facilitate devising the treatment strategy for PA patients and facilitate tailored postoperative follow-up scheduling.

### Limitations

Several limitations are present in this investigation. Despite the data being acquired from a selective and homogeneous population, the retrospective nature of this investigation suffered from a monocentric experience. Furthermore, other inflammatory markers such as cytokines, which may provide more profound details regarding the interaction between inflammation and proliferation, were not available in this retrospective investigation. Moreover, there are potential interlaboratory differences regarding the determination methods of pituitary hormone concentrations and also the determination methods of the MIB-1 labeling index in cancer tissue (e.g., digital imaging analysis, hotspot, average method) [51]. Those limitations must be considered before our results can be transferred to clinical practice or external validation. Therefore, a multicentric prospective trial with a thorough, homogeneously balanced study protocol should provide external validation for this score to enable its reliable integration into the healthcare of PA patients.

## 5. Conclusions

A strong association between the MIB-1 labeling index and the probability of PFS in completely resected pituitary adenomas was found. Moreover, we created a score (“FATE”), which may preoperatively facilitate a tailored estimation of the MIB-1 index which enhances the preoperative physician–patient dialog, PA surgery planning, and a thorough risk/benefit evaluation of the healthcare for pituitary adenoma patients.

## Figures and Tables

**Figure 1 jcm-11-07151-f001:**
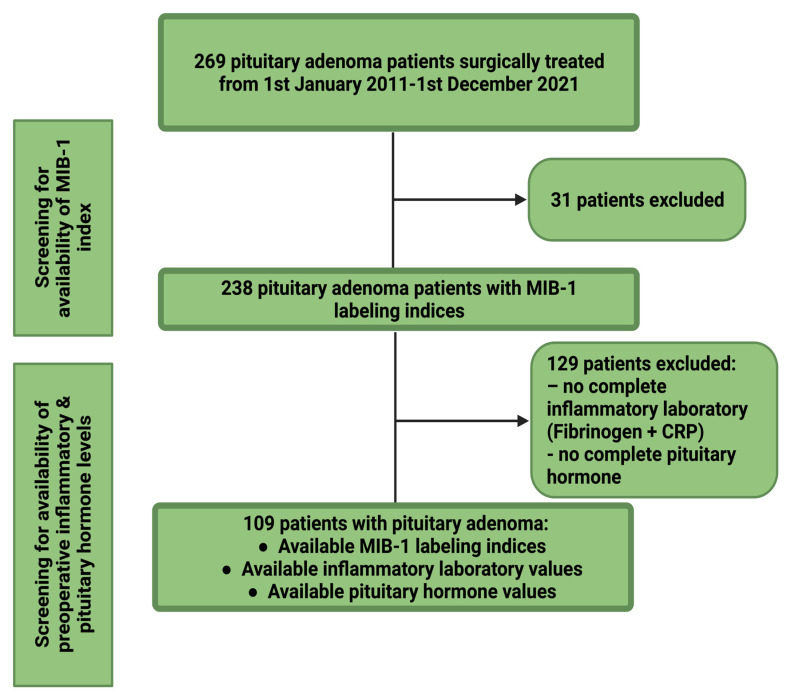
Flowchart summarizing the selection process of consecutive pituitary adenoma patients between 1 January 2011 and 1 December 2021. Abbreviations: The Molecular Immunology Borstel (MIB-1)-1 labeling index, c-reactive protein (CRP).

**Figure 2 jcm-11-07151-f002:**
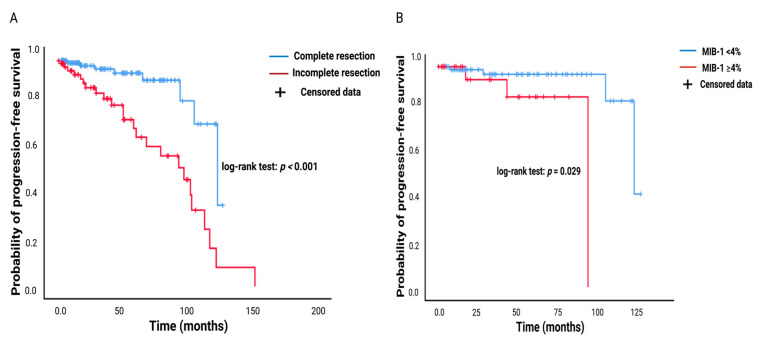
(**A**) Kaplan–Meier chart of tumor progression probability stratified by “complete resection” (blue line) and “incomplete resection” (red line). Censored patients are labeled by the vertical dashes (here: Absence of PA progression at last follow-up) within the progression-free survival curves. The time axis is right-censored at 200 months. *p* < 0.001 (log-rank test). (**B**) Kaplan–Meier chart of PA progression probability stratified by “MIB-1 ≥ 4%” (red line) and “MIB-1 < 4%” (blue line) in completely resected pituitary adenomas. The time axis is right-censored at 125 months. *p* = 0.029. Abbreviations: The Molecular Immunology Borstel (MIB-1)-1 labeling index.

**Figure 3 jcm-11-07151-f003:**
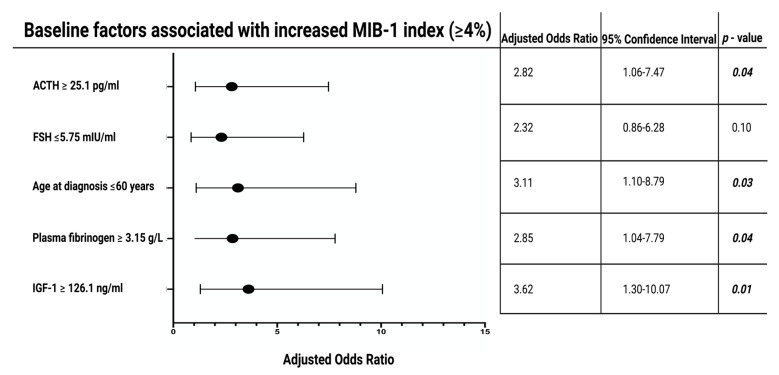
Forest plots illustrating the results of the multivariable binary logistic regression analysis: IGF-1 ≥ 126.1 ng/mL, age ≤ 60 years at diagnosis, ACTH ≥ 25.1 pg/mL, and plasma fibrinogen ≥ 3.15 g/L are variables being independently associated with an elevated MIB-1 labeling index. Black circles show the adjusted odds ratio for each variable and the corresponding lines represent the 95% confidence interval. *p*-values written in bold and italics label statistically significant results. Abbreviations: The Molecular Immunology Borstel (MIB-1)-1 labeling index, Adrenocorticotropic hormone (ACTH), Follicle-Stimulating Hormone (FSH), Insulin-like growth factor 1 (IGF-1).

**Figure 4 jcm-11-07151-f004:**
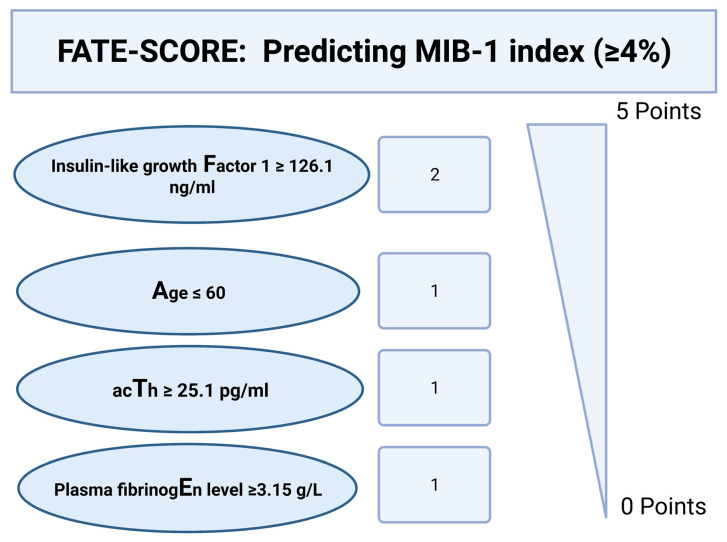
A preoperative clinical scoring system to identify patients at risk of an elevated MIB-1 labeling index (≥4%). A cumulative total score of <3 points results in an 85.2% probability of not having an elevated proliferative potential. Abbreviations: The Molecular Immunology Borstel (MIB-1)-1 labeling index, Adrenocorticotropic hormone (ACTH).

**Figure 5 jcm-11-07151-f005:**
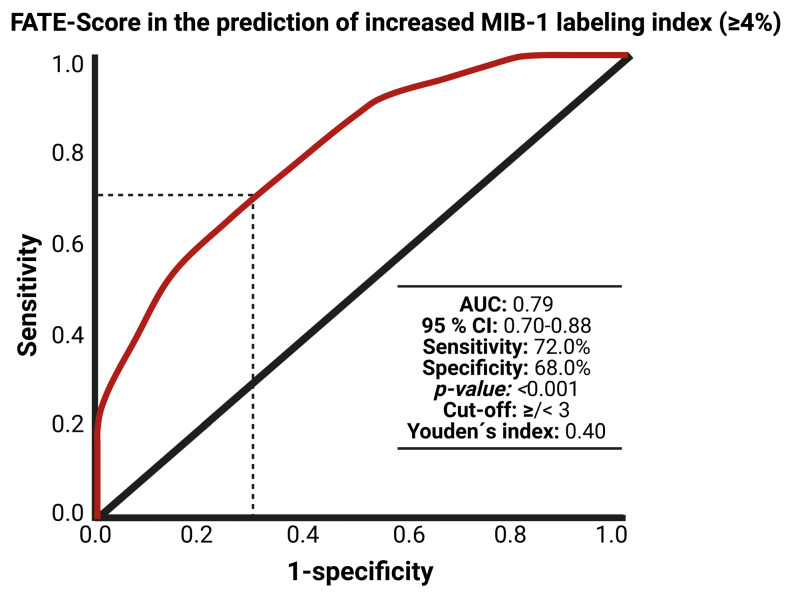
Receiver operating characteristic curve displaying FATE score in the preoperative estimation of the MIB-1 labeling index (≥4%). Abbreviations: The Molecular Immunology Borstel (MIB-1)-1 labeling index, area under the curve (AUC).

**Figure 6 jcm-11-07151-f006:**
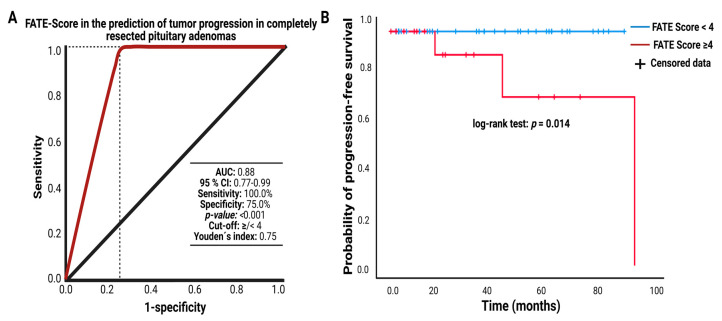
(**A**) Receiver operating characteristic curve displaying the FATE score in the prediction of tumor progression after a transsphenoidal GTR of a pituitary adenoma (**B**) Kaplan–Meier analysis of tumor progression probability stratified by “score: 0–3 points” (blue line) and “score: ≥ 4 points” (red line). Vertical dashes indicate censored data (here: progression-free at the last follow-up) within the progression-free survival curves. The time axis is right-censored at 100 months. *p* = 0.014 (log-rank test). Abbreviations: area under the curve (AUC).

**Table 1 jcm-11-07151-t001:** Patient characteristics (*n* = 109).

Median Age (IQR) (In y)	60 (47–70)
Sex Female Male	49 (45.0%) 60 (55.0%)
Median preoperative KPS (IQR)	90 (90–100)
Maximum diameter (mm)	21 (14–27)
Cavernous sinus invasion	43 (39.4%)
Pituitary apoplexy	6 (5.5%)
Extent of resection Gross total resection Subtotal resection	67 (61.5%) 42 (38.5%)
Pathological tumor types Gonadotroph adenoma Plurihormonal adenoma Null cell adenoma Corticotroph adenoma Somatotroph adenoma Prolactinoma Thyrotroph adenoma	49 (45%) 35 (32.1%) 11 (10.1%) 8 (7.3%) 3 (2.8%) 2 (1.8%) 1 (0.9%)
Median MIB-1 labeling index (%), IQR	3 (2–4)
p53 expression (available in 88 patients)	44 (50.0%)

Abbreviations: interquartile range (IQR), The Molecular Immunology Borstel (MIB-1)-1 labeling index.

**Table 2 jcm-11-07151-t002:** Preoperative demographic, clinical, imaging, and laboratory features in pituitary adenoma patients with a normal and increased MIB-1 labeling index. *P*-values written in italics and bold label statistically significant results. (*n* = 109).

Variable	MIB-1 < 4% (*n* = 77)	MIB-1 ≥ 4% (*n* = 32)	*p*-Value
Sex Female Male	38 (49.4%) 39 (50.6%)	11 (34.3%) 21 (65.5%)	0.21
Age, mean ± SD	61.4 ± 15.6	49.3 ± 15.9	<0.001
Preoperative KPS, mean ± SD	90.9 ± 10.8	92.2 ± 8.7	0.55
Body mass index, mean ± SD	28.7 ± 5.8	30.8 ± 9.1	0.16
Acromegaly Present Not present	12 65	6 26	0.78
Cushing´s disease Present Not present	2 75	2 30	0.58
Diabetes Present Not present	8 (10.4%) 69 (89.6%)	6 (18.8%) 26 (81.2%)	0.35
Acetylsalicylic acid intake Present Not present	8 10.4%) 69 (89.6%)	3 (18.8%) 29 (81.2%)	0.99
Pituitary apoplexy Present Not present	4 (5.2%) 73 (94.8%)	2 (6.3%) 30 (93.7%)	0.99
Maximum diameter (mm), mean ± SD	21.8 ± 9.8	21.3 ± 9.0	0.81
Microadenoma Macroadenoma	13 (16.9%) 64 (83.1%)	6 (18.7%) 26 (81.3%)	0.79
Plurihormonal (immunohistochemical) Present Not present	24 (31.2%) 53 (61.8%)	11 (34.4%) 21 (65.6%)	0.82
Cavernous sinus invasion Present Not present	28 (36.4%) 49 (63.6%)	15 (46.9%) 17 (53.1%)	0.39
Preoperative hormone replacement Hydrocortisone Levothyroxine Testosterone Levothyroxine and testosterone Hydrocortisone and levothyroxine Hydrocortisone and testosterone and levothyroxine None	12 (15.6%) 11 (14.3%) 0 (0.0%) 1 (1.3%) 10 (13.0%) 1 (1.3%) 42 (54.5%)	11 (34.4%) 5 (15.6%) 1 (3.1%) 0 (0.0%) 4 (12.5%) 0 (0.0%) 11 (34.4%)	0.14
p53 expression (available in 88 patients) present not present	27 (46.55%) 31 (53.45%)	17 (56.66%) 13 (43.33%)	0.50
Baseline Plasma fibrinogen (g/L), mean ± SD	3.19 ± 0.89	3.55 ± 0.89	0.06
Baseline Serum C-reactive protein (mg/I), mean ± SD	4.76 ± 14.81	6.22 ± 11.55	0.62
Baseline TSH (µU/mL), mean ± SD	1.30 ± 0.83	1.44 ± 0.81	0.45
Baseline IGF-1 (ng/mL), mean ± SD	189.56 ± 238.51	293.46 ± 310.77	0.06
Baseline ACTH (pg/mL), mean ± SD	26.76 ± 20.34	39.54 ± 34.36	0.03
Baseline LH (U/I), mean ± SD	5.15 ± 6.40	3.56 ± 4.68	0.22
Baseline FSH (mIU/mL), mean ± SD	13.53 ± 18.47	8.07 ± 12.63	0.09
Baseline prolactin (ng/mL), mean ± SD	29.18 ± 33.00	28.48 ± 39.73	0.93
Baseline estradiol (pg/mL), mean ± SD	21.38 ± 25.77	47.84 ± 93.66	0.38
Baseline testosterone (ng/mL), mean ± SD	1.98 ± 1.85	1.53 ± 1.11	0.20
Baseline growth hormone (ng/mL), mean ± SD	5.15 ± 12.64	6.50 ± 12.24	0.69

## Data Availability

All data is included in the present manuscript.

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
