# Peer review of "Preoperative Risk Stratification of Increased MIB-1 Labeling Index in Pituitary Adenoma: A Newly Proposed Prognostic Scoring System"

_jcm, 2022, doi:10.3390/jcm11237151_

Round 1
Reviewer 1 Report
This is an interesting proposal to evaluate pituitary adenomas with readily available laboratory tests and to make reliable predictions.
The authors evaluated several factors that are clinically relevant and found a combination that may be useful in everyday practice. Over time, additional cases and follow up will improve their data and may be amenable to verification by other authors and cohorts.
An important data missing is the percentage of micro and macroadenomas evaluated in each group.
I find it interesting that a large proportion of the tumors evaluated had a Ki-67 >4% (around 30%) and were positive for p53 or plurihormonal. All of these characteristics are also considered to be factors for progression in these tumors. The authors must clarify if they consider this findings to be frequent in their population or if they are related to the characteristics of the center or the patient selection, which could be understandable bias.
The finding of an elevated IGF above the proposed cutoff point appears to be independent of the presence of acromegaly. Apparently almost any patient with normal or elevated IGF-1 may hace this point in the score. I would think that this may be truth mainly for microadenomas that do not affect normal pituitary function or acromegaly. Was there a subanalysis performed using the normal ranges for sex and age?
Considering that these questions can be answered, I would consider the publication as optimal.
Author Response
Dear Reviewer
Thank you for reading our manuscript and critically reviewing it, which will help us improve it to a better scientific level and make it more understandable to the readership.
We agree with the reviewer that data regarding the proportion of micro- and macroadenomas among the MIB-1 labeling indices groups are very important. Hence, we have added this information to the section “3.4 Screening for associations between MIB-1 labeling index and patient characteristics”. Tumor size was not found to be associated with the MIB-1 labeling index. Thirteen patients (16.9%) with a MIB-1 index <4% had a microadenoma, and 6 patients (18.7%) with a MIB-1 index ≥4% had a microadenoma, respectively (p = 0.79). Furthermore, mean maximum diameter (in mm) in those with a MIB-1 index <4% was 21.8± 9.8, whereas mean maximum diameter in those patients with a MIB-1 index ≥4% was 21.3 ± 9.0 (p = 0.81).
The data have been also added to the table 2 in the section “3.4 Screening for associations between MIB-1 labeling index and patient characteristics”.
The reviewer is absolutely right that our institutional proportions of plurihormonal adenomas and p53 expression are high and might influence the MIB-1 labeling index. Therefore, we have added the analyses of potential associations between MIB-1 labeling index with p53 expression or plurihormonal PAs to the section “3.4 Screening for associations between MIB-1 labeling index and patient characteristics”. The expression of p53 was not found to be associated with the MIB-1 labeling index. P53 expression was homogeneously distributed among both MIB-1 labeling indices groups. Immunohistochemically detected plurihormonal PAs were observed in 24 cases (31.2%) among patients with a MIB-1 index <4%, whereas 11 plurihormonal PAs (34.4%) were found among those with a MIB-1 index ≥4% (Fisher´s exact test (two-sided): p = 0.82). The mean MIB-1 labeling index among those with a plurihormonal PA was 3.09 ± 1.74, and 3.18 ± 1.76 among those with a non-plurihormonal PA, respectively (p = 0.80).
Plurihormonal PAs were not found to be associated with an increased MIB-1 index in our cohort. However, it has to be reminded that the determination methods of plurihormonal PA using clinical signs, serum hormone concentrations, and pathological results are increasingly discussed [1]. In the present series, the plurihormonal PAs are determined using immunohistochemical methods as recommended by other investigators [1, 2]. Nevertheless, immunohistochemical detection of hormones might not always result in a laboratory increase of the serum hormone concentrations or in clinical endocrine signs. Hence, it is debatable whether this classification has some clinical implication. This phenomenon might be explained by the fact that hormones secreted by the PA are biologically inactive or they lost their functioning after the entry into the blood system [1, 3]. Moreover, in the present series we could not identify an association between p53 expression and an increased MIB-1 labeling index. The prognostic value of determining p53 expression in PAs is controversial and there is no recommendation to routinely included it in the classification workflow according to the WHO [4]. Furthermore, different staining methods and heterogenous cohorts (proportions of micro- and macroadenomas) resulted also in a broad range from 17% to 60% regarding the frequency of observed p53 expression [5, 6].
We agree with the reviewer that age or sex might influence the baseline IGF-1 values. Consequently, we have added the subanalyses investigating the associations of the variables age and sex with IGF-1 concentrations to the section “3.4 Screening for associations between MIB-1 labeling index and patient characteristics”. Twenty-seven (27/49; 55.1%) female patients had an IGF-1 ≥ 126.1 ng/ml, whereas 23 (23/60; 38.3%) male patients had an IGF-1 ≥ 126.1 ng/ml (Fisher´s exact test (two-sided): p = 0.09). Patients with an IGF-1 ≥ 126.1 ng/ml had a mean age of 50.4 ± 16.2, whereas those with an IGF-1 <126.1 ng/ml had a mean age of 64.2 ± 14.3 (independent t-test: p < 0.001). Therefore, the subanalyses identified an association between IGF-1 and age. The variable age was also included in our multivariable analysis and was identified as an independent predictor of MIB-1 labeling index.
References
- Wei, L.; Yue, Z.; Wang, S. Immunopathological study of plurihormonal pituitary adenomas. Chin J Neurosurg. 2008, 13, 208
- Ho, D.M.; Hsu, C.Y.; Ting, L.T.; Chiang, H. Plurihormonal pituitary adenomas: immunostaining of all pituitary hormones is mandatory for correct classification. Histopathology. 2001, 39, 310-9
- Shi, R.; Wan, X.; Yan, Z.; Tan, Z.; Liu, X.; Lei, T. Clinicopathological Characteristics of Plurihormonal Pituitary Adenoma. Front Surg. 2022, 9, 826720
- Louis, D.N.; Perry, A.; Reifenberger, G.; von Deimling, A.; Figarella-Branger, D.; Cavenee, W.K.; Ohgaki, H.; Wiestler, O.D.; Kleihues, P.; Ellison, D.W. The 2016 World Health Organization Classification of Tumors of the Central Nervous System: a summary. Acta Neuropathol. 2016, 131(6), 803-820
- Ozer, E.; Canda, M.S.; Ulukus, C.; Guray, M.; Erbayraktar, S. Expression of Bcl-2, Bax and p53 proteins in pituitary adenomas: an immunohistochemical study. Tumori. 2003, 89(1), 54-59
- Zakir, J.C.; Casulari, L.A.; Rosa, J.W.; Rosa, J.W.; de Mello, P.A.; de Malhaes, A.V.; Naves, L.A. Prognostic Value of Invasion, Markers of Proliferation, and Classification of Giant Pituitary Tumors, in a Georeferred Cohort in Brazil of 50 Patients, with a Long-Term Postoperative Follow-Up. Int J Endocrinol. 2016, 2016, 7964523
Reviewer 2 Report
I read with interest the retrospective review entitled “Preoperative risk stratification of increased MIB-1 labeling index in pituitary adenoma: a newly proposed prognostic scoring system”. The authors are to be commended on their review. Notably, they found that patients with elevated IGF-I, ACTH, Fibrinogen and younger age were more likely to have elevated MIB-1 (>4%) which may be related to more aggressive tumors. The authors furthermore propose a scoring mechanism using these facts to predict elevated MIB-1 levels preoperatively. The concept is novel and has some value. The data on the predictability of aggressiveness by MIB-1 is mixed and knowing that a tumor that has an elevated MIB-1 preoperatively might change how the patient is managed, namely these patients might not undergo surgery and be treated upfront with medical or radiation treatments. Although interesting, the practicality and generalizability of this study is difficult. The authors describe recurrence as growth of over 25% of the original tumor which is quite a lot of growth. Lab values can be quite variable from center to center and population to population.
Author Response
Dear Reviewer
Thank you for reading our manuscript and critically reviewing it, which will help us improve it to a better scientific level and make it more understandable to the readership.
In the following we would like to respond to your remarks:
The reviewer is absolutely right that the definition of a progression as an 25 % increase in the volume of the previous pituitary tumor size can be quite a lot [1]. We agree with the reviewer that the introduction of a new treatment for recurrence might not be completely dependent on the increase of volume. For instance, in case of a new regrowth with a short distance to the optic chiasm most physicians will not strictly adhere to this defined cut-off (25%) regarding the term progression. However, in the present series we wanted to analyze the prognostic value of the MIB-1 index regarding the regrowth. Hence, we had to adhere to a strictly determined imaging definition of PA progression to enable a reliable analysis of the role of MIB-1 index in PA progression.
Moreover, we agree with the reviewer that there are potential interlaboratory differences regarding the determination methods of pituitary hormone concentrations or also the determination methods of the MIB-1 labeling index in cancer tissue (e.g., digital imaging analysis, hotspot, average method) [2]. Those limitations must be considered before our results are transferred to the clinical practice or an external validation is provided. We added this important information to the section “4.1 Limitations” in the discussion.
References
- Gerges, M.M.; Rumalla, K.; Godil, S.S.; Younus, I.; Elshamy, W.; Dobri, G.A.; Kacker, A.; Tabee, A.; Anand, V.K.; Schwartz, T.H. Long-term outcomes after endoscopic endonasal surgery for nonfunctioning pituitary macroadenomas. J Neurosurg. 2020, Jan 31, 1-12
- Jang, M.H.; Kim, H.J.; Chung, Y.R.; Lee, Y.; Park, S.Y. A comparison of Ki-67 counting methods in luminal Breast Cancer: The Average Method vs. the Hot Spot Method. PLoS One. 2017, 12(2), e0172031